# OpenReview forum: "Red Teaming Language-Conditioned Robot Models via Vision Language Models"
_NeurIPS.cc/2024/Workshop/SafeGenAi — SafeGenAi Poster_

### Official Review · Reviewer_Sewr · 2024-10-09
**Solid paper, but not inspiring enough for a workshop presentation**

**Rating:** 5
**Confidence:** 4

**Review:**

### Summary
The paper introduces a novel evaluation framework for language-conditioned robot models using VLMs to generate diverse and challenging instructions. ERT aims to assess the safety and effectiveness of these models beyond conventional benchmarks by creating instructions that test the robot's ability to handle complex and unforeseen scenarios. However, the paper itself is more like prompt engineering, which might not be able to facilitate the workshop's discussion board.

### Strengths
- ERT introduces a systematic approach to generate and test instructions that reflect real-world complexities.
- By focusing on safety and the ability to handle novel scenarios, ERT addresses critical gaps in current robotic model evaluations.
- The integration of vision and language processing to generate context-specific instructions is a notable technical advancement.
- The paper provides really comprehensive appendix of the prompts (nearly 100 pages), which is of great effort.

### Weaknesses
- The need for sophisticated setups involving VLMs and dynamic instruction generation could complicate the adoption of ERT.
- While effective, the current implementation of ERT is primarily tested on manipulation tasks, and its applicability to other types of robotic operations remains to be explored.

---

### Official Review · Reviewer_CbJp · 2024-10-10
**The paper aligns well with the workshop theme, using VLMs to generate challenging instructions for stress testing robot models. However, the approach lacks originality/signifcance, as LLMs and VLMs have already been widely used for similar tasks. Due to this, I am rating the paper as marginally below the acceptance threshold.**

**Rating:** 5
**Confidence:** 4

**Review:**

**Strength**

_Alignment with workshop:_

The paper aligns really well with the theme of the workshop. The paper proposes the use of VLM to generate challenging instructions for stress testing language-conditioned robot models. The paper also refines the generated challenging instructions by adding the previously generated challenging instructions to the VLM prompt.

_Writing style and clarity:_

The paper is well written and all the details are clearly laid out.

---

**Weakness**

_Originality and Significance:_
The biggest weakness of the paper is its significance. The proposed approach is not new. LLM's (including VLM's) have been used to generate synthetic data for wide range of applications like model training, stress testing. LLM's (auto raters) are also used for evaluating the output of other LLM's. Therefore I feel the contributions in this paper are not very significant.

Because of this drawback I am rating this paper 5 (Marginally below acceptance threshold)

---

### Official Review · Reviewer_4seu · 2024-10-10
**Embodied Red Teaming Framework for VLM-Guided Robots  - Good analysis with room for improvement**

**Rating:** 7
**Confidence:** 3

**Review:**

Review Overview:\
This work introduces the Embodied Red Teaming framework to test the capabilities and safety of existing VLM-guided robots. The framework is novel and the authors present a good analysis; however, the existing experiments could be strengthened and more experiments could bolster the argument.\
This review follows the sections of the paper. Strengths (+) and weaknesses (-) are noted for each section.

Introduction:\
\+ Well-written, clear intro. Good examples in lines 34-35, 53, and 64.\
\- Lines 41-42: Is there an example of a method and dataset where the score drops that significantly? Such an experiment would strengthen this work's motivation.\
\- Red teaming bears resemblance to the concept of adversarial attacks in machine learning. Is this the same as red teaming? A brief statement of their similarity or difference would be informative as most readers are aware of "adversarial attacks" in ML.

Preliminaries:\
Embodied Red Teaming:\
\- In Equation 1, what's R? A reward function? Later it is called a metric function; maybe move that description up here where Equation 1 is defined.

Experiments:\
\- The naive baseline is a good idea, but the work could benefit from more such baselines. For example, randomly adding and deleting a word to give a noisy instruction, substituting synonyms for the main verb or noun, etc.\
\- Table 1 would be more meaningful if the rows contained the same task. So ERT was for the same tasks as the CALVIN or RLBench, then we could see how ERT differs in its wordings. Right now, it is hard to tell whether ERT phrasing is more challenging because the reader doesn't know what the original prompt was. Maybe make two separate tables side by side, one comparing CALVIN and ERT and the other RLBench and ERT. Adding a rephrase column would also be helpful!\
\- The diversity scores experiment is good; however, in Figure 3, I would keep the scales of the 3 plots the same. Right now at first glance, it looks like the difference between the green bar and blue bar is about the same in the BLEU and CLIP plot, which is misleading. Because of the scales, it's actually a big difference which the author correctly points out in the caption.

Discussion and Analysis:\
\- The results in line 322 or figures a and b are not very interesting, as without fine-tuning the robot is of course expected to follow any instructions given. It would have been more interesting if you attempt to fine-tune for this and see if the robot rejects unsafe instructions or if there are still any unsafe instructions that can be found that it would perform (maybe adapt ERT to provide unsafe instructions instead of a more difficult phrasing).\
\- For Line 366 figures c and d, how are these neutral instructions found? If they were part of one of the datasets, that would make this point more impactful. Characterizing how often a neutral instruction is unsafe within the dataset would be ideal.\
\- The example around line 387 is making me rethink whether ERT is too difficult. If someone asked me to "Suppress the glow" of an LED, I probably would not think to turn it off, but I would point it in the other direction or put a bedsheet over it or something to make it dimmer. It would be interesting to do a user study of ERT and see if the users can actually complete the task correctly given the task phrasing.\
\- I wonder if the length of the instruction has something to do with the poor performance. It seems like the ERT instructions tend to be longer, even in Table 1. Maybe this is an "ablation" that should be explored.\
\+ The idea of "embodied similarity" is a very sound conclusion to draw. There is room for more explanation and more experiments to prove it though, other than these qualitative observations of the ERT outputs. Maybe ask GPT to produce instructions that are "human-centric" and then measure how well that performs.

Conclusion:\
\+ Good structure, addressing interested audiences and presenting limitations concisely.